# Association of organizational and patient behaviors with physician well-being: A national survey in China

Xiaoyu Wang[1], Yimei Zhu[2], Fang Wang[3], Yuan Liang[1]*

1 Department of Social Medicine and Health Management, School of Public Health, Tongji Medical College, Huazhong University of Science and Technology, Wuhan, China, 2 School of Media, Communication and Sociology, University of Leicester, Leicester, United Kingdom, 3 Department of Epidemiology and Health Statistics, School of Health, Wuhan University, Wuhan, China

* liangyuan217@hust.edu.cn, liangyuan217@163.com

**Data Availability Statement:** All relevant data are within the paper and its Supporting Information files.

**Funding:** This study was supported by grants from National Natural Science Foundation of China (:

## Abstract

This study aims to investigate the association of organizational and patient behaviors (reflecting the internal and external environment of hospital, respectively) with physician well-being. A national cross-sectional survey was conducted in 77 hospitals across seven provinces in China between July 2014 and April 2015. Physician well-being was assessed with job satisfaction, career regret and happiness. Organizational behaviors were assessed with organizational fairness, leadership attention and team interaction; patient behaviors were assessed with patient trust and unreasonable requests from patients. Of a study sample of 3,159 physicians, 1,788 were men (56.6%) and 1,371 were women (43.4%). Overall, positive organizational and patient behaviors reported by physicians were relatively low. Negative organizational behaviors and patient behaviors including lower organizational fairness, lower leadership attention, lower team interaction and lower patient trust were associated with lower job satisfaction and lower life satisfaction, and higher career regret. The association between organizational behaviors and physician well-being exhibited some gender differences, while no clear gender difference was found for the relationship between patient behaviors and physician well-being. Given the importance of physician well-being for the healthcare system, interventions for improving internal and external hospital environments (e.g., organizational fairness, leadership attention, team interaction and patient trust) may benefit physician well-being.

## Introduction

Physician well-being is particularly important in the healthcare system, including its impact on physicians' own physical and mental health and career development [1–5], as well as the quality of medical services and patient satisfaction [6–9]. Studies on improving physician well-being revealed various underlying factors that may provide benefits. While some studies have emphasized the importance of individual intrinsic motivational factors [10–12], others have suggested that [6, 13], faced with widespread physicians' distress, individual susceptibility

71273098). The funders had no role in study design, data collection and analysis, decision to publish, or preparation of the manuscript.

**Competing interests:** The authors have declared that no competing interests exist.

(individual factors) may not be primary, and environmental factors (including the internal and external hospital environments) may be more important.

Most research on the internal hospital environments has focused on working environment factors [14–19], including workload, workflow, job autonomy, hospital teaching attributes, practice characteristics, scheduling issues, leadership behaviors, use of electronic health records, number of doctors (or nurses) per ward bed, average daily admissions, and average occupancy workload. In fact, factors such as job autonomy, hospital teaching attributes, practice characteristics, and scheduling issues are difficult for healthcare providers (including clinicians and hospitals/organizations) to modify [20, 21] because the internal hospital environments are broad in scope and relatively complex to examine. Compared to the structural factors of organization, organizational behaviors (e.g., organizational fairness, leadership attention, and team interaction) may be more feasible to modify. In terms of the external hospital environment including the social environment, previous research [22, 23] has focused on financial compensation, and external competition, and less on the attitudes and behaviors of patients and their families. Although previous studies have incorporated patient factors into working conditions for physicians and hospitals [6,10], patients are situated in a societal context and are affected by social environmental factors outside the hospital. Physicians have closer contact with patients in their everyday work environment than those who they receive incentives or penalties from such as medical insurance agencies; hence factors related to patients may be more relevant to improve physician well-being. In addition, some previous studies [24–27] have focused on the associations between leadership behaviors and physicians' burnout and satisfaction, however, the types of organizational behaviors are relatively simple [24, 25, 27, 28], such as leadership and/or organization fairness, and few explanatory variables were accounted for, lacking some organizational characteristics (e.g., specialties, teaching nature, and the ratio of physicians to beds) and physicians' family factors, etc.

The present study used data from a national survey of physicians in China to explore the association of organizational and patient behaviors (reflecting the internal and external environment of hospital, respectively) with physician well-being.

## Methods

### Study design and participants

We conducted a survey between July 2014 and April 2015 using stratified cluster sampling strategy. Briefly, seven provinces were selected from each geographical area, with two from each of East and West China (Shandong and Jiangsu, and Gansu and Yunnan), and one each from South, Central and North China (Guangdong, Hubei and Beijing metropolis). The details of this survey have been described in a previous report [29], and there was a total of 85 eligible hospitals in the selected regions with 77 hospitals (90.60%) agreeing to participate. In each hospital, convenience sampling was used to select four surgical departments and four internal medicine departments (excluding obstetrics and pediatrics). A total of 528 departments were selected and all of their full-time physicians (n = 5,754) were eligible and invited to participate in the survey. Trained survey interviewers sent copies of the questionnaire) to each department, along with an explanation of the survey purpose and method. After one or two days, the interviewers returned to collect completed questionnaires. Participation was voluntary, and all data was kept confidential. Finally, 4,281 (74.4% response rate) physicians responded to the survey and 634 (11.0%) invalid questionnaires were excluded. The final valid responses were 3,159 (54.9%) after excluding 488 (8.5%) responses with incomplete information (S1 Fig in S1 File).

All participants provided oral informed consent for interviews. We obtained ethical approval from the institutional review board at the Tongji Medical College, Huazhong University of Science and Technology (Wuhan, China) [No. IORG0003571].

### Data collection

Exposure factors include three specific aspects of organizational behaviors (organizational fairness, leadership attention, team interaction) and two aspects of patient behaviors (patient trust and unreasonable requests from patients). One of the commonly used measurement for organizational behaviors is Colquitt's Organizational Justice Scale [30, 31], which contains 4 dimensions and 20 items, although the length of this questionnaire limits its feasibility in nationwide studies. Based on the literature and expert consultation [32–36], we adopted similar survey indicators and methods in the current study. Organizational fairness was assessed using two single-item measures adapted from the full Colquitt's Organizational Fairness Scale: pay fairness (reflecting distributional fairness), and task fairness (reflecting procedural fairness). Similarly, leadership attention was assessed using two single-item measures: interests attention (reflecting attention to physicians' material needs) and opinions attention (reflecting attention to physicians' emotional needs). Team interaction [37] was assessed using two single-item measures: number of dinners with colleagues per month (reflecting social interaction) and number of clinical case discussions with colleagues per month (reflecting work interaction). Unlike some prior studies [38–40], patient behaviors in this study were measured from the physicians' perspective as two items of patient trust (intrinsic behavior) and unreasonable requests from patients (explicit behavior) were used as measures for patient behaviors. Each question was answered on a 5-point Likert scale and we recoded each question into four categories (see Table 2).

There were three outcome variables: job satisfaction, career regret and happiness. In reference to prior studies [11], three outcome variables were assessed using three single-item measures respectively in the current study. Job satisfaction was examined using the question "Overall, how would you rate your satisfaction with your work?". Career regret was examined with the question "If you had an opportunity to choose your profession, would you become a physician again?" and the response was reverse coded to measure the degree of "regret". Happiness was assessed with the question "Overall, what do you think your happiness score is?" and the response for happiness was reverse-coded as categorical variables (80–100 = "very high", 60–79 = "higher", 40–59 = "average", 20–39 = "lower" and 0–19 = "very low"). Each question was answered using a 5-point Likert scale with response options ranging from "very low" to "very high". Finally, all responses for the three outcome variables were recoded as binary variables: low ("very low/lower/average") versus high ("higher/very high").

In addition, we measured a number of factors previously reported to be associated with well-being of physicians, including socio-demographics (age, gender, marital status, education level, economic status and professional ranking), hospitals and department characteristics (hospital level, hospital type, academic status, physician specialty and the ratio of physicians to beds) and family support. Data of hospitals and department characteristics was obtained from the unit heads.

### Statistical analysis

To adjust for nonresponses, the data was weighted by respondents' age and gender, according to the hospitals' demographic information issued by the National General Hospital in 2015.

For crude comparisons, we used chi-square tests or Kruskal-Wallis tests for categorical variables and used binary logistic regression models to examine the association of organizational

and patient behaviors with physician well-being. Given that women are considered having more family care responsibility whilst work obligations negatively impact family care roles [41, 42], there may be a gender difference for the impact of organizational and patient behaviors on physician well-being; hence the analysis of this study did a gender stratification. Two-sided tests were used for all the analyses, and *P* values of 0.05 or less were considered statistically significant. All analyses were performed using SPSS, version 22.0 (SPSS Inc., Chicago, IL, USA).

## Results

The primary characteristics of participants, departments and hospitals were summarized in Table 1. Among 3,159 respondents, 1788 were men (56.6%) and 1,371 were women (43.4%). More than 80% of physicians were married and less than 10% reported good financial conditions. In terms of gender difference, females were less likely to be surgeons than males (vs 25.8% vs 63.3%, *P*<0.001).

Table 2 showed the characteristics of physicians' self-perceived organizational and patient behaviors. Overall, the rates of positive responses regarding organizational and patient behaviors reported by physicians were relatively low. Less than 10% of physicians reported good pay fairness and large interests attention from leaders. More than 70% of physicians had only 0–1 dinner with colleagues per month and 30–40% of physicians had ≥4 clinical case discussions with colleagues per month. Less than 15% reported high patient trust. Concerning gender difference, females were more likely to report fewer dinner with colleagues per month than males (70.4% vs 79.7%, *P*<0.001) and clinical case discussions (19.7% vs 29.0%, *P*<0.001).

Table 3 reported gender differences on job satisfaction, career regret and level of happiness. More than half of physicians felt high level of happiness, however, physicians were not optimistic about their work, with low job satisfaction (30%–40%) and high career regret (nearly 70%). Additionally, differences in level of happiness between men and women were observed. Females reported higher level of happiness than males(54.6% vs 51.9%, *P*<0.05).

Table 4 summarized the association of organizational and patient behaviors with physician well-being after adjusting for all other explanatory variables. In general, patient behaviors were more significantly associated with physician well-being than organizational behaviors. In terms of patient behaviors, lower patient trust was associated with lower job satisfaction (OR $_{total}$ = 0.22, 95% CI: 0.15–0.33, OR $_{men}$ = 0.21, 95% CI: 0.12–0.36, OR $_{women}$ = 0.18, 95% CI: 0.09–0.37) and lower level of happiness (OR $_{total}$ = 0.32, 95% CI: 0.21–0.49, OR $_{men}$ = 0.34, 95% CI: 0.19–0.58, OR $_{women}$ = 0.22, 95% CI: 0.10–0.48) and higher career regret (OR $_{total}$ = 5.32, 95% CI: 3.34–8.46, OR $_{men}$ = 4.99, 95% CI: 2.75–9.07, OR $_{women}$ = 8.20, 95% CI: 3.68–18.24). With regards to organizational behaviors, lower pay fairness was positively associated with higher career regret (OR $_{total}$ = 2.80, 95% CI: 1.74–4.51, OR $_{men}$ = 2.85, 95% CI: 1.48–5.49, OR $_{women}$ = 3.07, 95% CI: 1.44–6.54); lower task fairness was positively associated with lower level of happiness (OR $_{total}$ = 0.43, 95% CI: 0.28–0.67, OR $_{men}$ = 0.42, 95% CI: 0.23–0.75, OR $_{women}$ = 0.41, 95% CI: 0.21–0.82) and more frequent clinical case discussions with colleagues per month was positively associated with higher job satisfaction (OR $_{total}$ = 1.46, 95% CI: 1.15–1.86, OR $_{men}$ = 1.43, 95% CI: 1.02–1.99, OR $_{women}$ = 1.68, 95% CI: 1.15–2.46).

In addition, significant gender differences were observed in the relationship between some organizational behaviors and physician well-being. Specifically, females who reported lower leadership attention to interests were more likely to have lower job satisfaction (OR $_{women}$ = 0.20, 95% CI: 0.09–0.42) and higher career regret (OR $_{women}$ = 4.73, 95% CI: 2.03–11.04). Male physicians who reported lower leadership attention to opinions were more likely to have higher career regret (OR $_{men}$ = 2.29, 95% CI: 1.24–4.24). Having dinners with colleagues three times per month was associated with higher job satisfaction (OR $_{women}$ = 2.28, 95% CI: 1.07–

**Table 1. The primary characteristics of participants, hospitals and departments.**

| | Men(n = 1788) | Women(n = 1371) | *P* Value |
|---|---|---|---|
| **Socio-demographics** | | | |
| Age, y | | | |
| ≤34 | 519(29.0) | 398(29.0) | <0.001 |
| 35–44 | 620(34.7) | 476(34.7) | |
| ≥45 | 649(36.3) | 498(36.3) | |
| Marital status | | | |
| Single/Other | 252(14.7) | 254(19.1) | 0.001 |
| Married | 1466(85.3) | 1076(80.9) | |
| Education level | | | |
| Undergraduate and below | 893(50.7) | 739(55.9) | 0.010 |
| Master | 636(36.1) | 464(35.1) | |
| Doctor | 234(13.3) | 120(9.1) | |
| Economic status | | | |
| Very poor | 227(12.7) | 183(13.4) | 0.889 |
| Poor | 392(22.0) | 251(18.4) | |
| Average | 1011(56.6) | 828(60.6) | |
| Good | 155(8.7) | 105(7.7) | |
| Professional ranking | | | |
| Primary/Other | 374(23.0) | 300(25.6) | <0.001 |
| Intermediate | 490(30.2) | 397(33.8) | |
| Senior | 761(46.8) | 477(40.6) | |
| **Hospitals and departments characteristics** | | | |
| Hospital level | | | |
| Secondary Hospital | 255(14.3) | 207(15.1) | 0.509 |
| Tertiary Hospital | 1533(85.7) | 1164(84.9) | |
| Hospital type | | | |
| Traditional Chinese Medicine | 470(26.3) | 413(30.1) | 0.017 |
| Western Medicine | 1318(73.7) | 958(69.9) | |
| Academic status | | | |
| No | 1402(78.4) | 1134(82.7) | 0.003 |
| Yes | 386(21.6) | 237(17.3) | |
| Physician specialty | | | |
| Internal medicine | 656(36.7) | 1018(74.3) | <0.001 |
| Surgery | 1132(63.3) | 353(25.8) | |
| The ratio of physicians to beds | | | |
| <0.20 | 473(26.8) | 389(28.9) | 0.020 |
| 0.20–0.30 | 777(44.0) | 485(36.1) | |
| ≥0.30 | 517(29.3) | 471(35.0) | |
| **Family support** | | | |
| Very small/Small | 71(4.0) | 37(2.7) | 0.023 |
| Average | 330(18.5) | 215(15.7) | |
| Large/Very large | 1384(77.5) | 1118(81.6) | |

4.85) and higher level pf happiness (OR $_{women}$ = 2.57, 95% CI: 1.10–6.02) of women physicians, while having dinners with colleagues ≥4 times was associated with lower career regret (OR $_{men}$ = 0.49, 95% CI: 0.32–0.77) and higher level of happiness (OR $_{men}$ = 2.07, 95% CI: 1.32–3.25) of men physicians.

**Table 2. The characteristics of physicians' self-perceived organizational and patient behaviors.**

| | Men(n = 1788) | Women(n = 1371) | P Value |
|---|---|---|---|
| Organizational behaviors | | | |
| **Organizational fairness** | | | |
| Pay fairness | | | |
| Very poor | 533(29.8) | 359(26.2) | <0.001 |
| Poor | 504(28.2) | 374(27.3) | |
| Average | 619(34.6) | 512(37.4) | |
| Good/Very good | 131(7.3) | 124(9.1) | |
| Task fairness | | | |
| Very poor | 295(16.5) | 188(13.7) | 0.002 |
| Poor | 388(21.7) | 295(21.5) | |
| Average | 877(49.1) | 696(50.8) | |
| Good/Very good | 227(12.7) | 191(13.9) | |
| **Leadership attention** | | | |
| Interests attention | | | |
| Very small | 595(33.3) | 409(29.8) | 0.009 |
| Smalle | 413(23.1) | 334(24.4) | |
| Average | 606(33.9) | 513(37.4) | |
| Large/Very large | 172(9.6) | 115(8.4) | |
| Opinions attention | | | |
| Very small | 634(35.5) | 439(32.0) | 0.020 |
| Smalle | 419(23.5) | 337(24.6) | |
| Average | 542(30.3) | 485(35.4) | |
| Large/Very large | 192(10.7) | 110(8.0) | |
| **Team interaction** | | | |
| The number of dinners with colleagues per month | | | |
| ≥4times | 147(8.3) | 80(5.9) | <0.001 |
| 3times | 101(5.7) | 53(3.9) | |
| 2times | 279(15.76) | 145(10.6) | |
| 0-1time | 1255(70.4) | 1090(79.7) | |
| The number of clinical case discussions per month | | | |
| ≥4times | 831(46.7) | 460(33.8) | <0.001 |
| 3times | 273(15.3) | 215(15.8) | |
| 2times | 327(18.4) | 290(21.3) | |
| 0-1time | 350(19.7) | 395(29.0) | |
| Patient behaviors | | | |
| **Patient trust** | | | |
| Very low | 267(15.0) | 174(12.8) | 0.642 |
| Lower | 497(27.9) | 409(30.1) | |
| Average | 801(45.0) | 645(47.4) | |
| Higher/Very high | 214(12.0) | 133(9.8) | |
| **Unreasonable requests from patients** | | | |
| A lot | 112(6.3) | 80(5.8) | 0.031 |
| More | 477(26.7) | 412(30.1) | |
| Average | 601(33.7) | 470(34.3) | |
| Less/Rarely | 595(33.3) | 408(29.8) | |

**Table 3. Job satisfaction, career regret and happiness of physicians by gender.**

|  | Men(n = 1788) | Women(n = 1371) | P Value |
|---|---|---|---|
| **Job Satisfaction** |  |  |  |
| Very low | 157(8.8) | 93(6.8) | 0.314 |
| Lower | 176(9.8) | 153(11.2) |  |
| Average | 810(45.3) | 601(43.8) |  |
| Higher | 506(28.3) | 372(27.1) |  |
| Very High | 139(7.8) | 152(11.1) |  |
| **Career Regret** |  |  |  |
| Very low | 61(3.4) | 36(2.6) |  |
| Lower | 142(7.9) | 134(9.8) | 0.686 |
| Average | 394(22.0) | 288(21.0) |  |
| Higher | 322(18.0) | 254(18.5) |  |
| Very High | 869(48.6) | 659(48.1) |  |
| **Happiness** |  |  |  |
| Very low | 198(11.1) | 130(9.5) |  |
| Lower | 230(12.9) | 141(10.3) | 0.002 |
| Average | 432(24.2) | 352(25.7) |  |
| Higher | 684(38.3) | 517(37.7) |  |
| Very High | 244(13.7) | 231(16.9) |  |

## Discussion

To our knowledge, this study is the first study of physician well-being and its relevance to organizational and patient behaviors using a national representative sample in China. In general, more negative organizational behaviors and patient behaviors were positively associated with lower job satisfaction, lower level of happiness and higher career regret. Poor well-being and negative organizational and patient behaviors are common issues that confronts both male and female physicians in China, despite slight gender differences in various aspects.

These findings are relevant to current clinical practice and can apply to other countries, given the prevailing focus on physician well-being internationally. The current results confirm that the physician well-being should be approached from the perspective of internal and external environments. Whether organizational behaviors of the internal environment or patient behaviors of the external environment, healthcare providers (including hospital organizations and clinicians) can take action to change these behaviors, which can be modified to modify than structural factors of organization. The distribution of the exposure factors revealed that the proportion of positive responses was relatively low for both organizational and patient behaviors. Combined with the association of organizational and patient behaviors (reflecting the internal and external environment of hospital, respectively) with physician well-being, the current findings suggest that addressing these factors could substantially improve physician well-being.

Previous research [43, 44] indicates that low levels of organizational fairness can make employees feel excluded, potentially explaining the association of organizational fairness with physicians' job satisfaction, level of happiness and career regret in this study. Specially, lower pay fairness was positively associated with higher career regret, and lower task fairness was positively associated with lower level of happiness. Prior work [45] indicates that, compared with distribution fairness [46] (reflected by pay fairness in this study), procedural fairness (reflected by task fairness in this study) may have a significant impact on people's psychological health, attitudes and values, which is consistent with the results of this study.

**Table 4. Multivariable logistic regression results for association of organizational and patient behaviors-related effects for physicians well-being.**

| | Job Satisfaction | | | Career Regret | | | Happiness | | |
|---|---|---|---|---|---|---|---|---|---|
| | Total OR(95% CI) | Men OR (95% CI) | Women OR 95%CI) | Total OR(95% CI) | Men OR(95% CI) | Women OR (95%CI) | Total OR(95% CI) | Men OR(95% CI) | WomenOR (95%CI) |
| Organizational behaviors | | | | | | | | | |
| **Organizational fairness** | | | | | | | | | |
| Pay fairness | | | | | | | | | |
| Very poor | 1.44(0.91,2.28) | 1.25(0.66,2.37) | 1.53(0.75,3.13) | 2.80 (1.74,4.51)*** | 2.85 (1.48,5.49)** | 3.07 (1.44,6.54)** | 0.88(0.54,1.44) | 0.64 (0.32,1.26) | 1.41(0.67,2.96) |
| Poor | 0.91(0.59,1.40) | 0.86(0.47,1.57) | 0.92(0.47,1.83) | 2.85 (1.84,4.39)*** | 2.13 (1.19,3.83)* | 4.70 (2.30,9.58)*** | 0.88(0.55,1.40) | 0.61 (0.32,1.15) | 1.35(0.67,2.75) |
| Average | 1.13(0.75,1.69) | 1.49(0.85,2.60) | 0.76(0.40,1.44) | 1.99 (1.34,2.98)*** | 1.46 (0.85,2.50) | 3.16 (1.62,6.15)*** | 0.88(0.57,1.36) | 0.74 (0.40,1.35) | 1.00(0.51,1.98) |
| Good/ Very good | 1.00 | 1.00 | 1.00 | 1.00 | 1.00 | 1.00 | 1.00 | 1.00 | 1.00 |
| Task fairness | | | | | | | | | |
| Very poor | 0.50 (0.33,0.75)*** | 0.36 (0.21,0.63)*** | 0.71(0.36,1.41) | 1.24(0.77,2.00) | 1.24 (0.65,2.34) | 1.11 (0.50,2.46) | 0.43 (0.28,0.67)*** | 0.42 (0.23,0.75)** | 0.41 (0.21,0.82)* |
| Poor | 0.68 (0.47,0.91)*** | 0.65(0.40,1.05) | 0.66(0.36,1.20) | 1.09(0.75,1.59) | 1.33 (0.80,2.19) | 0.79(0.41,1.51) | 0.52 (0.36,0.76)*** | 0.53 (0.33,0.88)* | 0.49 (0.27,0.87)* |
| Average | 0.60 (0.44,0.83)*** | 0.47 (0.31,0.71)*** | 0.86(0.51,1.44) | 0.98(0.71,1.35) | 1.21 (0.79,1.84) | 0.65(0.37,1.13) | 0.73(0.52,1.02) | 0.64 (0.41,1.00)* | 0.89(0.53,1.52) |
| Good/ Very good | 1.00 | 1.00 | 1.00 | 1.00 | 1.00 | 1.00 | 1.00 | 1.00 | 1.00 |
| **Leadership attention** | | | | | | | | | |
| Interests attention | | | | | | | | | |
| Very small | 0.43 (0.27,0.67)*** | 0.66(0.37,1.18) | 0.20 (0.09,0.42)*** | 1.54(0.95,2.49) | 0.73 (0.38,1.38) | 4.73 (2.03,11.04)*** | 0.74(0.46,1.18) | 0.80 (0.43,1.48) | 0.67(0.31,1.47) |
| Smalle | 0.46 (0.31,0.70)*** | 0.57 (0.33,0.98)* | 0.29 (0.14,0.60)*** | 1.76(1.16,2.69) | 0.89 (0.51,1.56) | 5.06 (2.37,10.77)*** | 0.83(0.54,1.28) | 1.00 (0.57,1.76) | 0.62(0.30,1.30) |
| Average | 0.55 (0.38,0.79)*** | 0.71(0.44,1.14) | 0.34 (0.18,0.65)*** | 1.20(0.83,1.74) | 0.66 (0.41,1.08) | 3.41 (1.74,6.67)*** | 0.87(0.59,1.29) | 0.94 (0.57,1.56) | 0.76(0.39,1.48) |
| Large/ Very large | 1.00 | 1.00 | 1.00 | 1.00 | 1.00 | 1.00 | 1.00 | 1.00 | 1.00 |
| Opinions attention | | | | | | | | | |
| Very small | 1.19(0.75,1.87) | 1.05(0.58,1.87) | 1.29(0.59,2.84) | 1.35(0.84,2.18) | 2.29 (1.24,4.24)** | 0.73(0.30,1.74) | 0.86(0.53,1.37) | 0.95 (0.52,1.75) | 0.62(0.27,1.40) |
| Smalle | 1.09(0.71,1.66) | 1.18(0.69,2.03) | 0.83(0.39,1.74) | 0.92(0.60,1.40) | 1.35 (0.78,2.31) | 0.52(0.24,1.14) | 1.09(0.70,1.70) | 1.16 (0.66,2.03) | 0.87(0.40,1.89) |
| Average | 1.35(0.92,1.97) | 1.30(0.81,2.10) | 1.22(0.62,2.39) | 0.80(0.65,1.17) | 1.04 (0.65,1.67) | 0.51(0.25,1.04) | 1.03(0.69,1.55) | 1.21 (0.73,2.00) | 0.74(0.36,1.52) |
| Large/ Very large | 1.00 | 1.00 | 1.00 | 1.00 | 1.00 | 1.00 | 1.00 | 1.00 | 1.00 |
| **Team interaction** | | | | | | | | | |
| The number of dinners with colleagues per month | | | | | | | | | |
| ≥4times | 0.69 (0.48,1.00)** | 0.95(0.61,1.48) | 0.37 (0.18,0.77)** | 0.58 (0.40,0.83)*** | 0.49 (0.32,0.77)** | 0.78(0.39,1.56) | 1.48 (1.03,2.14)*** | 2.07 (1.32,3.25)** | 0.67(0.35,1.30) |
| 3times | 1.30(0.86,1.97) | 1.05(0.62,1.76) | 2.28(1.07,4.85) | 0.89(0.58,1.35) | 1.01 (0.60,1.69) | 0.65(0.31,1.38) | 1.89 (1.21,2.95)*** | 1.86 (1.08,3.21)* | 2.57 (1.10,6.02)* |
| 2times | 0.80(0.61,1.05) | 0.95(0.68,1.32) | 0.61(0.36,1.02) | 0.96(0.73,1.26) | 0.99 (0.70,1.40) | 0.89(0.54,1.47) | 1.35 (1.03,1.76)*** | 1.26 (0.90,1.75) | 1.57(0.96,2.57) |
| 0-1time | 1.00 | 1.00 | 1.00 | 1.00 | 1.00 | 1.00 | 1.00 | 1.00 | 1.00 |

*(Continued)*

**Table 4.** (Continued)

| | Job Satisfaction | | | Career Regret | | | Happiness | | |
|---|---|---|---|---|---|---|---|---|---|
| | Total OR(95% CI) | Men OR (95% CI) | Women OR 95%CI) | Total OR(95% CI) | Men OR(95% CI) | Women OR (95%CI) | Total OR(95% CI) | Men OR(95% CI) | WomenOR (95%CI) |
| The number of clinical case discussions per month | | | | | | | | | |
| ≥4times | 1.46 (1.15,1.86)*** | 1.43 (1.02,1.99)* | 1.68 (1.15,2.46)** | 0.98(0.76,1.27) | 1.13 (0.80,1.61) | 0.85(0.57,1.27) | 0.93(0.73,1.18) | 0.85 (0.61,1.19) | 1.08(0.74,1.56) |
| 3times | 1.10(0.82,1.47) | 1.42(0.94,2.12) | 0.88(0.55,1.42) | 1.11(0.81,1.52) | 1.31 (0.85,2.03) | 0.88(0.54,1.45) | 0.89(0.66,1.20) | 0.78 (0.52,1.18) | 1.10(0.70,1.73) |
| 2times | 0.90(0.68,1.20) | 0.83(0.56,1.24) | 1.03(0.67,1.57) | 1.07(0.80,1.43) | 0.97 (0.65,1.45) | 1.20(0.75,1.91) | 0.96(0.73,1.26) | 1.06 (0.72,1.55) | 0.86(0.57,1.31) |
| 0-1time | 1.00 | 1.00 | 1.00 | 1.00 | 1.00 | 1.00 | 1.00 | 1.00 | 1.00 |
| Patient behaviors | | | | | | | | | |
| **Patient trust** | | | | | | | | | |
| Very low | 0.22 (0.15,0.33)*** | 0.21 (0.12,0.36)*** | 0.18 (0.09,0.37)*** | 5.32 (3.34,8.46)*** | 4.99 (2.75,9.07)*** | 8.20 (3.68,18.24)*** | 0.32 (0.21,0.49)*** | 0.34 (0.19,0.58)*** | 0.22 (0.10,0.48)*** |
| Lower | 0.23 (0.17,0.33)*** | 0.20 (0.13,0.31)*** | 0.25 (0.14,0.45)*** | 3.95 (2.80,5.58)*** | 3.81 (2.45,5.94)*** | 5.28 (2.89,9.64)*** | 0.48 (0.33,0.69)*** | 0.55 (0.35,0.86)** | 0.33 (0.17,0.64)** |
| Average | 0.37 (0.27,0.51)*** | 0.31 (0.21,0.46)*** | 0.39 (0.22,0.67)*** | 1.89 (1.39,2.57)*** | 1.93 (1.30,2.85)*** | 2.36 (1.36,4.09)** | 0.70 (0.50,0.99)*** | 0.72 (0.47,1.11) | 0.55(0.29,1.06) |
| Higher/ Very high | 1.00 | 1.00 | 1.00 | 1.00 | 1.00 | 1.00 | 1.00 | 1.00 | 1.00 |
| **Unreasonable requests from patients** | | | | | | | | | |
| A lot | 0.84(0.54,1.30) | 0.88(0.48,1.60) | 0.81(0.41,1.60) | 3.37 (1.70,6.69)*** | 3.31 (1.40,7.82)** | 2.78(0.86,8.97) | 0.52 (0.32,0.84)*** | 0.52 (0.26,1.02) | 0.44 (0.21,0.92)* |
| More | 0.70 (0.55,0.90)*** | 0.71 (0.51,0.99)* | 0.62 (0.41,0.92)* | 1.53 (1.18,1.98)*** | 2.04 (1.44,2.91)*** | 1.14(0.75,1.72) | 0.58 (0.45,0.73)*** | 0.64 (0.46,0.88)** | 0.45 (0.30,0.66)*** |
| Average | 0.82(0.66,1.02) | 0.83(0.62,1.11) | 0.79(0.55,1.14) | 1.21(0.96,1.52) | 1.21 (0.91,1.63) | 1.27(0.87,1.86) | 0.73 (0.58,0.91)*** | 0.89 (0.66,1.19) | 0.54 (0.37,0.78)*** |
| Less/ Rarely | 1.00 | 1.00 | 1.00 | 1.00 | 1.00 | 1.00 | 1.00 | 1.00 | 1.00 |

Note: All Models for total participants adjusted for socio-demographics (gender, age, marital status, education level, economic status and professional ranking), hospitals and department characteristics (hospital level, hospital type, academic status, physician specialty and the ratio of physicians to beds) and family support. The stratified analysis adjusted for all the covariates except for the stratified variable

* $p < 0.05$

** $p < 0.01$

*** $p < 0.001$

In general, the mechanisms underlying the association between leadership attention and physician well-being by gender were less clear. Specially, lower leadership attention to interests was positively associated with lower job satisfaction and higher career regret of female physicians, and lower leadership attention to opinions was positively associated with higher career regret of male physicians. This gender difference may be related to traditional cultural difference between men and woman that men are concerned about their career while women are more concerned about their family responsivities. Additional studies will be required to explore this issue in more depth in future.

It may be unsurprising that more frequent clinical case discussions with colleagues per month was positively associated with higher job satisfaction which indicated, the important

role of case discussions on professional development, which is not related to gender [47, 48]. However, having more dinners with colleagues per month were beneficial to both male and female physicians. Specially, having dinners with colleagues 3 times per month was associated with higher job satisfaction and higher level of happiness of women physicians, while having dinners with colleagues ≥4 times was associated with lower career regret and higher happiness of men physicians. This finding may have occurred because women tend to carry greater family responsibilities than men in China, having high-intensity social interaction may place a disproportionate burden on women [49, 50].

This study confirmed the important role of patient trust as there was a positive association between patient trust and physician well-being which was consistent with findings from previous studies [51–53].

Notably, China's health care systems are based on public hospitals, and they follow the principles of cost control, efficiency improvement and patient-centered service, which are the same for countries with different health care systems around the world. Therefore, the lessons learnt from this study in relation to organizational behavior and patient behavior of public hospitals may be applicable to other countries.

As a strength of our study, the current analysis combined multiple organizational and patient behaviors, and the data was adjusted for a broad array of hospital organization and physicians' demographic characteristics and family support factors known to influence physicians' well-being. Importantly, our study involved several limitations. First, although we conducted a national survey, our study design did not enable us to identify differences between responding and non-responding physicians. Accordingly, the data was weighted by respondents' age and gender to adjust for nonresponses. Second, data was self-reported by physicians. Thus, both underreporting and overreporting may have occurred, with the potential for variation in overreporting and underreporting regarding organizational and patient behaviors, as well as physician well-being, introducing uncertainty regarding the generalizability of our study findings. Third, given the busyness of participants' job and the feasibility in a nationwide survey, the outcome variables in this study were measured using single-item indicators [5, 11, 32], and the measurement bias was possible. Fourth, the observational cross-sectional nature of the analyses precluded the ability to draw causal inferences. Nonetheless, the current study provided a large sample and multivariate evidence for in-depth investigation, which may enlighten future research.

## Conclusions

The national sample investigated physician well-being and its relevance to organizational behaviors and patient behaviors in China. In general, more negative organizational and patient behaviors (e.g., organizational fairness, leadership attention, team interaction and patient trust) were positively associated with lower physician well-being. The association between organizational behaviors and physician well-being exhibited some gender differences, while no clear gender difference was found for the relationship between patient behaviors and physician well-being. Given the prevailing international research interest in distress among physicians, interventions from the perspective of the internal and external hospital environment may provide a valuable approach for improving physician well-being.

## Supporting information

**S1 File.**
(DOCX)

## Acknowledgments

We would like to thank all the health staff and inpatients at 77 general hospitals who participated in the study. We thank Benjamin Knight, MSc., from Liwen Bianji, Edanz Editing China (www.liwenbianji.cn/ac), for editing the English text of a draft of this manuscript.

## Author Contributions

**Conceptualization:** Xiaoyu Wang, Yimei Zhu, Fang Wang, Yuan Liang.

**Data curation:** Yuan Liang.

**Formal analysis:** Xiaoyu Wang, Yuan Liang.

**Funding acquisition:** Yuan Liang.

**Investigation:** Xiaoyu Wang, Fang Wang, Yuan Liang.

**Methodology:** Xiaoyu Wang, Yimei Zhu, Fang Wang, Yuan Liang.

**Project administration:** Yuan Liang.

**Software:** Xiaoyu Wang.

**Supervision:** Yuan Liang.

**Validation:** Xiaoyu Wang.

**Visualization:** Xiaoyu Wang, Yimei Zhu.

**Writing – original draft:** Xiaoyu Wang, Yimei Zhu, Fang Wang, Yuan Liang.

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
