## [Decision Letter · Decision Letter 0]

1 Apr 2021

PONE-D-20-31349

Association of Organizational and Patient Behaviors with Physician Well-being: A National Survey in China

PLOS ONE

Dear Dr. Liang,

Thank you for submitting your manuscript to PLOS ONE. After careful consideration, we feel that it has merit but does not fully meet PLOS ONE’s publication criteria as it currently stands. Therefore, we invite you to submit a revised version of the manuscript that addresses the points raised during the review process.

The manuscript has been evaluated by two reviewers, and their comments are available below. Both reviewers have raised concerns and the manuscript will need significant revision before it can be considered for publication – you should anticipate that the reviewers will be re-invited to assess the revised manuscript, so please ensure that your revision is thorough. I have outlined some of the key concerns noted by the reviewers below, but you should respond to all concerns mentioned by the reviewers in your response-to-reviewers document. 

The key concerns noted by the reviewers relate to the assessment tools and their underlying constructs, as well as the rationale for stratifying all analyses by gender. These issues impact the interpretation of the results and should be explored.

We look forward to receiving your revised manuscript.

Kind regards,

Danielle Poole

Staff Editor

PLOS ONE

Journal Requirements:

2. In the methods section please provide additional information regarding the following:

-    Please include additional information regarding the survey or questionnaire content validation.  

-    Please provide a justification for the sample size used in your study, including any relevant power calculations (if applicable

-    Details regarding the participant eligibility criteria

-    A description on how participants were recruited for the study

Furthermore, if the questionnaire is not under a copyright more restrictive than CC-BY, please include a copy, in both the original language and English, as Supporting Information.

Reviewers' comments:

Reviewer's Responses to Questions

**Comments to the Author**

1. Is the manuscript technically sound, and do the data support the conclusions?

Reviewer #1: Partly

Reviewer #2: Yes

2. Has the statistical analysis been performed appropriately and rigorously? 

Reviewer #1: Yes

Reviewer #2: Yes

3. Have the authors made all data underlying the findings in their manuscript fully available?

Reviewer #1: Yes

Reviewer #2: Yes

4. Is the manuscript presented in an intelligible fashion and written in standard English?

Reviewer #1: No

Reviewer #2: Yes

5. Review Comments to the Author

Reviewer #1: I have several concerns with the assessment tools utilized in the study

Overall, what do you think your happiness score is? – these are validated QoL measures that should have been used if the authors were interested in evaluating QoL

If you had an opportunity to choose your profession, would you become a physician again – this does not measure intention to leave/turnover but rather career regret

Leadership scores could have been assessed using validated tools

Team interaction being assessed by number of times that teams go out to dinner is highly subjective and likely not generalizable.

Unclear how patient behavior was assessed but these findings go against prior research (patient expectations and gender disparities in medicine).

Unclear how pay fairness in China is generalizable to other countries given the profoundly different pay structure

Reviewer #2: This study used a national physician survey in China to examine the association of organizational characteristics and patient behaviors with physician well-being. The authors found that organizational characteristics were substantially associated with physician job satisfaction, turnover intention, and life happiness.

[1] Major

1. Pages 6-7. Statistical analysis and results

The authors conducted all analysis stratified by physician gender.

(1) Please clarify the logic underlying why the authors stratified the entire analyses by physician gender. For example, I was wondering if the authors assume physician gender is the most critical component that influence or differentiate physician well-being. Although some estimation results were different by physician gender, differences could be observed even when the authors stratified the analysis by other factors such as physician specialty, hospital level (secondary vs. tertiary), or hospital type (traditional Chinese medicine vs. western medicine).

(2) If the table space allows, I recommend presenting estimation results among the total study population (in which physician gender and all explanatory variables are adjusted) and stratifying by physician gender.

2. Page 7, fourth paragraph

I recommend deleting this paragraph and the associated results in the appendix table for two reasons:

(1) The authors have three co-primary outcomes (job satisfaction, turnover intention, life happiness). In their "sensitivity analysis," the same variable was used as an outcome variable in one estimation, but as an explanatory variable in another estimation (e.g., job satisfaction is an outcome variable, then becomes an explanatory variable for turnover intention). In statistics, this implies simultaneity between two outcomes, which would require a more advanced estimation model to generate unbiased estimates.

(2) I am not sure whether “sensitivity analysis” usually refers to the adjustment of additional explanatory variable(s) while all other parts of the empirical design (e.g., applicable study population, categorization of outcome/independent variable) are the same.

[2] Minor

1. P.6. Statistical analysis

The authors stated, “Data were weighted to adjust for non-responses so that physician responding to the initial questions were matched to the demographics of the total hospitalized population issued by the National General Hospital.” Please clarify whether this means that adjusting weight made the data nationally representative at the physician level.

2. P.7. Results section

(1) Please include the 95% confidence interval for each odds ratio (OR).

(2) Please specify whether the difference in association of organizational/patient characteristics with outcomes between men and women is a difference in magnitudes (i.e., OR) or statistical significance. Indeed, estimates show that some associations are statistically significant among men but not among women, and vice versa.

3. Please clarify some terminologies in the text. I have two examples:

(1) Page 7, second paragraph. The authors stated, “Table 4 …. after adjustment for potentially confounding factors.” I suggest replacing “potentially confounding factors” with “all other explanatory variables” because the term “confounding factors” usually means unobserved factors included in the error term, which are correlated with independent variables and cause bias in estimates.

(2) Page 7, fourth paragraph. I am not sure whether “sensitivity analysis” is the correct terminology in this context.

6. PLOS authors have the option to publish the peer review history of their article (what does this mean?). If published, this will include your full peer review and any attached files.

Reviewer #1: No

Reviewer #2: No

---

## [Author Response · Author response to Decision Letter 0]

8 May 2021

please see the file named with "Response to Reviewers 2021-4-27"

---

## [Decision Letter · Decision Letter 1]

7 Jun 2021

PONE-D-20-31349R1

Association of Organizational and Patient Behaviors with Physician Well-being: A National Survey in China

PLOS ONE

Dear Dr. Liang, 

Thank you for submitting your manuscript to PLOS ONE. After careful consideration, we feel that it has merit but does not fully meet PLOS ONE’s publication criteria as it currently stands. Therefore, we invite you to submit a revised version of the manuscript that addresses the points raised during the review process.

In addition to the reviewers’ comments, please note a few points:

(1) The reviewers were concerned that some of the responses to the previous comments were not complete. Specifically, the discussion of the limitations in methods (e.g., validity of the variables) requires a clearer statement.

(2) Please briefly discuss similarities and differences in (a) health care systems between China and other countries, and (b) study outcomes and policy implications from your study and relevant literature from other countries. This would be helpful to emphasize your contribution to the literature.

(3) The reviewer recommended including interaction terms between gender and other explanatory variables.

(a) I would recommend adding an interaction term between gender and one primary explanatory variable (e.g., hospital teaching status) per multivariable regression and checking whether the interaction term is statistically significant.

(b) Repeat (a) for a few explanatory variables that the authors believe are most important.

(c) If no interaction term is statistically significant, please summarize the results in a few sentences.

(d) If any interaction term is statistically significant, please

(d.i) derive adjusted predicted outcome, average marginal effect, and difference-in-differences given that interpretation of interaction term in a non-linear model is different from that in a linear model, and

(d.ii) report them in a few sentences (or the authors may want to write an appendix table).

We look forward to receiving your revised manuscript.

Kind regards,

Hyo Jung Tak, PhD

Academic Editor

PLOS ONE

Reviewers' comments:

Reviewer's Responses to Questions

**Comments to the Author**

1. If the authors have adequately addressed your comments raised in a previous round of review and you feel that this manuscript is now acceptable for publication, you may indicate that here to bypass the “Comments to the Author” section, enter your conflict of interest statement in the “Confidential to Editor” section, and submit your "Accept" recommendation.

Reviewer #3: (No Response)

2. Is the manuscript technically sound, and do the data support the conclusions?

Reviewer #3: Partly

3. Has the statistical analysis been performed appropriately and rigorously? 

Reviewer #3: Yes

4. Have the authors made all data underlying the findings in their manuscript fully available?

Reviewer #3: Yes

5. Is the manuscript presented in an intelligible fashion and written in standard English?

Reviewer #3: Yes

6. Review Comments to the Author

Reviewer #3: Thank you for the opportunity to review your manuscript "Association of Organizational and Patient Behaviors with Physician Well-being: A National Survey in China". The physician's well-being is important for physicians themselves and the quality of care to patients. Please see my comments below. Thank you.

Reviewer#1

1. Comment 1 - This is a limitation of the study by not using validated measures of QoL.

2. Comment 4 has not been well addressed by the authors. The author-cited reference does not support the authors’ choice of the variable “number of times that teams go out to dinner”. The literature talks about various meetings in an organization, such as regular problem-solving team meetings, managerial meetings, or focus group meetings. These meetings are to facilitate interactions between team members and leadership in order to improve communication and coordination among members, and these activities are likely taking place in their regular work settings.

3. Comment 5 could be another limitation of the study as it did not capture patients’ perspectives directly from patients.

4. Comment 6 – Given China-specific data and context, the findings are likely limited for other countries.

Reviewer#2

1. [2] Minor Comment 1 – The author’s response is still vague whether the estimation in the study is nationally representative by incorporating survey design features, such as stratification,…… Please clarify it.

Additional comments

Abstract

1. Given the study has been conducted in one country, the findings may not be generalizable to other countries, and so caution should be taken with stating policy implications.

Introduction

1. As the study is about physician well-being, not distress, the 1st sentence in the 1st paragraph could divert readers from the focus. It could be relocated elsewhere or removed.

2. In the 2nd paragraph, the authors limited the working environment factors to those (job autonomy, workload, scheduling…) and separated organizational behaviors (organizational fairness, team interaction, and so on). Whether they are job-specific factors or organizational behaviors, both constitute a working environment. I would suggest making sure consistency with the literature in using and defining these languages.

3. In the same paragraph, a sentence starting with “Although previous studies have incorporated….”. Please cite a reference to support authors’ argument.

4. The next sentence starting with “compared with macro financial policy,..”. What is macro financial policy? And the whole sentence is not clear. Please rephrase it.

5. The sentence “types of organizational behaviors are relatively simple” needs supporting evidence.

6. Given this study is specific in China, more discussion of the background/context would be useful to help readers get better understanding. Also, I would suggest briefly discussing potential or expected contributions out of the study.

Data collection

1. In measuring team interaction, is there any justification/validation for including ‘number of dinners with colleagues per month’? This is quite subjective.

Statistical analysis

1. Were survey design features accounted for in the analyses to make it nationally representative? If so, please state it.

2. When it comes to gender difference, what authors did seems to be a sub-group analysis within each gender. If authors intended to compare genders in the outcomes of interest as seen throughout the manuscript, I would recommend using an interaction effect using gender with other explanatory variables.

Results

1. In the 3rd paragraph, P-value in the text is different from the one in Table.

2. It would be more appropriate to use ‘association’ than ‘correlates’ for the title of Table 4.

3. In Table 4, please add statistical significance, such as using p-values. And, revise “for physicians” to “physician well-being”

4. Throughout the manuscript, be consistent by using “gender” to indicate men or women.

Discussion

1. In the 1st paragraph, the 2nd sentence can be separated, one for the overall association and the other one by gender.

2. Delete current from “current findings”, and the phrase “which can be modified to some extent” is unclear.

3. In the 3rd paragraph, the 1st sentence would be part of study limitations “, and it is unclear about what the “psychological state” means and how the study findings are relevant to it.

4. In the 4th paragraph, the sentence starting with “It may be related….” seems to be not solely based on the study findings. For example, what authors say “women may be more concerned about their family” is not convincing.

5. In the 5th paragraph, please cite relevant literature as necessary.

6. In the 6th paragraph, ‘association’ than ‘correlation’ would be a better term and be consistent across the manuscript.

7. In the 7th paragraph, “Compared” is inappropriate since it does not compare between the number of unreasonable requests from patients and patient trust. It could confuse readers. And, is there any justification for the interpretation in “A possible explanation…”? And the following sentence “patients’ distrust is not related to their medical knowledge” is unclear and unconvincing. Also, “excellent techniques” itself is confusing.

7. PLOS authors have the option to publish the peer review history of their article (what does this mean?). If published, this will include your full peer review and any attached files.

Reviewer #3: No

---

## [Author Response · Author response to Decision Letter 1]

31 Oct 2021

Please view the uploaded files (named "PONE-S-20-39067 Response to Reviewers 2021-11-1") directly.

---

## [Decision Letter · Decision Letter 2]

30 Dec 2021

PONE-D-20-31349R2

Association of Organizational and Patient Behaviors with Physician Well-being: A National Survey in China

PLOS ONE

Dear Dr. Liang,

Thank you for submitting your manuscript to PLOS ONE. After careful consideration, we feel that it has merit but does not fully meet PLOS ONE’s publication criteria as it currently stands. Therefore, we invite you to submit a revised version of the manuscript that addresses the points raised during the review process.

We look forward to receiving your revised manuscript.

Kind regards,

Hyo Jung Tak, PhD

Academic Editor

PLOS ONE

Journal Requirements:

Additional Editor Comments (if provided):

[1] Major comments. None

[2] Minor comments. Please make sure all terms and numbers are consistent throughout the abstract, text, and tables. For percentages, please write the first decimal point only (e.g., 11.02% - > 11.0%; both in text and tables).

Reviewers' comments:

Reviewer's Responses to Questions

**Comments to the Author**

1. If the authors have adequately addressed your comments raised in a previous round of review and you feel that this manuscript is now acceptable for publication, you may indicate that here to bypass the “Comments to the Author” section, enter your conflict of interest statement in the “Confidential to Editor” section, and submit your "Accept" recommendation.

Reviewer #3: All comments have been addressed

2. Is the manuscript technically sound, and do the data support the conclusions?

Reviewer #3: Yes

3. Has the statistical analysis been performed appropriately and rigorously? 

Reviewer #3: Yes

4. Have the authors made all data underlying the findings in their manuscript fully available?

Reviewer #3: Yes

5. Is the manuscript presented in an intelligible fashion and written in standard English?

Reviewer #3: Yes

6. Review Comments to the Author

Reviewer #3: Comments to authors

I appreciate the revisions from authors.

Minor

1. Table. Interaction

I would suggest modifying the wording in Table for readability. For example, Gender*Pay fairness -> Men*Pay fairness.

2. The statement “Therefore, the study of organizational behavior and patient behavior of public hospitals in china may be universally applicable to other countries.”: This seems too generalizing the study findings. As the authors noted, the study effectively focuses on “organizational behavior” and “patient behavior” in public hospitals in one country. The findings could be relevant to some extent but should be interpreted with caution when it comes to different settings for example in other countries with healthcare systems that have a unique payment system, policy, regulation, culture,…...

7. PLOS authors have the option to publish the peer review history of their article (what does this mean?). If published, this will include your full peer review and any attached files.

Reviewer #3: No

---

## [Author Response · Author response to Decision Letter 2]

26 Feb 2022

Response to the editor’s and reviewers’ comments

（February 13, 2020）

PONE-D-20-31349R2

Association of Organizational and Patient Behaviors with Physician Well-being: A National Survey in China

Journal Requirements:

Response: We thank the editor for this comment and reminding.

We did not cite papers that have been retracted in our manuscript.

Additional Editor Comments (if provided):

[1] Major comments. None

[2] Minor comments. Please make sure all terms and numbers are consistent throughout the abstract, text, and tables. For percentages, please write the first decimal point only (e.g., 11.02% - > 11.0%; both in text and tables).

Response: We thank the reviewers for this kind reminders, and we have done as suggested.

Reviewers' comments:

Comments to the Author

1. If the authors have adequately addressed your comments raised in a previous round of review and you feel that this manuscript is now acceptable for publication, you may indicate that here to bypass the “Comments to the Author” section, enter your conflict of interest statement in the “Confidential to Editor” section, and submit your "Accept" recommendation.

Reviewer #3: All comments have been addressed

2. Is the manuscript technically sound, and do the data support the conclusions?

Reviewer #3: Yes

3. Has the statistical analysis been performed appropriately and rigorously?

Reviewer #3: Yes

4. Have the authors made all data underlying the findings in their manuscript fully available?

Reviewer #3: Yes

5. Is the manuscript presented in an intelligible fashion and written in standard English?

Reviewer #3: Yes

Response: We thank the reviewer for the positive comments.

Reviewer #3:Comments to authors

I appreciate the revisions from authors.

Minor

1. Table. Interaction

I would suggest modifying the wording in Table for readability. For example, Gender*Pay fairness -> Men*Pay fairness.

Response: We thank the reviewer for this kind reminders, and we have revised the wording in Table of the last response (November 1, 2021).

2. The statement “Therefore, the study of organizational behavior and patient behavior of public hospitals in China may be universally applicable to other countries.”: This seems too generalizing the study findings. As the authors noted, the study effectively focuses on “organizational behavior” and “patient behavior” in public hospitals in one country. The findings could be relevant to some extent but should be interpreted with caution when it comes to different settings for example in other countries with healthcare systems that have a unique payment system, policy, regulation, culture,…...

Response: We thank the reviewer for this kind reminders, and we have revised as following:

The original

Therefore, the study of organizational behavior and patient behavior of public hospitals in China may be universally applicable to other countries.

The revised 

Therefore, the study of organizational behavior and patient behavior of public hospitals in China may be partially applicable to other countries..

3. 7. PLOS authors have the option to publish the peer review history of their article (what does this mean?). If published, this will include your full peer review and any attached files.

Do you want your identity to be public for this peer review? For information about this choice, including consent withdrawal, please see our Privacy Policy.

Reviewer #3: No

Response: We thank the reviewer for the constructive comments.

---

## [Decision Letter · Decision Letter 3]

27 Apr 2022

Association of Organizational and Patient Behaviors with Physician Well-being: A National Survey in China

PONE-D-20-31349R3

Dear Dr. Liang,

We’re pleased to inform you that your manuscript has been judged scientifically suitable for publication and will be formally accepted for publication once it meets all outstanding technical requirements.

Kind regards,

Hyo Jung Tak, PhD

Guest Editor

PLOS ONE

Additional Editor Comments (optional):

Reviewers' comments:

Reviewer's Responses to Questions

**Comments to the Author**

1. If the authors have adequately addressed your comments raised in a previous round of review and you feel that this manuscript is now acceptable for publication, you may indicate that here to bypass the “Comments to the Author” section, enter your conflict of interest statement in the “Confidential to Editor” section, and submit your "Accept" recommendation.

Reviewer #3: All comments have been addressed

2. Is the manuscript technically sound, and do the data support the conclusions?

Reviewer #3: Yes

3. Has the statistical analysis been performed appropriately and rigorously? 

Reviewer #3: Yes

4. Have the authors made all data underlying the findings in their manuscript fully available?

Reviewer #3: Yes

5. Is the manuscript presented in an intelligible fashion and written in standard English?

Reviewer #3: Yes

6. Review Comments to the Author

Reviewer #3: (No Response)

7. PLOS authors have the option to publish the peer review history of their article (what does this mean?). If published, this will include your full peer review and any attached files.

Reviewer #3: No

---

## [Editor Report · Acceptance letter]

20 May 2022

PONE-D-20-31349R3 

Association of Organizational and Patient Behaviors with Physician Well-being: A National Survey in China 

Dear Dr. Liang:

I'm pleased to inform you that your manuscript has been deemed suitable for publication in PLOS ONE. Congratulations! Your manuscript is now with our production department. 

Kind regards, 

on behalf of

Dr. Hyo Jung Tak 

Guest Editor

PLOS ONE